# Differential Effect of Free-Air CO_2_ Enrichment (FACE) in Different Organs and Growth Stages of Two Cultivars of Durum Wheat

**DOI:** 10.3390/plants12030686

**Published:** 2023-02-03

**Authors:** Angie L. Gámez, Xue Han, Iker Aranjuelo

**Affiliations:** 1Agrobiotechnology Institute (IdAB), CSIC—Government of Navarre, 31192 Mutilva Baja, Spain; 2NAFOSA Company, 31350 Peralta, Spain; 3Institute of Environment and Sustainable Development in Agriculture, Chinese Academy of Agricultural Sciences (IEDA, CAAS), Beijing 100081, China

**Keywords:** free-air CO_2_ enrichment (FACE), phenology, plant acclimation, source, sink, wheat

## Abstract

Wheat is a target crop within the food security context. The responses of wheat plants under elevated concentrations of CO_2_ (*e*[CO_2_]) have been previously studied; however, few of these studies have evaluated several organs at different phenological stages simultaneously under free-air CO_2_ enrichment (FACE) conditions. The main objective of this study was to evaluate the effect of *e*[CO_2_] in two cultivars of wheat (Triumph and Norin), analyzed at three phenological stages (elongation, anthesis, and maturation) and in different organs at each stage, under FACE conditions. Agronomic, biomass, physiological, and carbon (C) and nitrogen (N) dynamics were examined in both ambient CO_2_ (*a*[CO_2_]) fixed at 415 µmol mol^−1^ CO_2_ and *e*[CO_2_] at 550 µmol mol^−1^ CO_2_. We found minimal effect of *e*[CO_2_] compared to *a*[CO_2_] on agronomic and biomass parameters. Also, while exposure to 550 µmol mol^−1^ CO_2_ increased the photosynthetic rate of CO_2_ assimilation (An), the current study showed a diminishment in the maximum carboxylation (*V_c,max_*) and maximum electron transport (*J_max_*) under *e*[CO_2_] conditions compared to *a*[CO_2_] at physiological level in both cultivars. However, even if no significant differences were detected between cultivars on photosynthetic machinery, differential responses between cultivars were detected in C and N dynamics at *e*[CO_2_]. Triumph showed starch accumulation in most organs during anthesis and maturation, but a decline in N content was observed. Contrastingly, in Norin, a decrease in starch content during the three stages and an increase in N content was observed. The amino acid content decreased in grain and shells at maturation in both cultivars, which might indicate a minimal translocation from source to sink organs. These results suggest a greater acclimation to *e*[CO_2_] enrichment in Triumph than Norin, because both the elongation stage and *e*[CO_2_] modified the source–sink relationship. According to the differences between cultivars, future studies should be performed to test genetic variation under FACE technology and explore the potential of cultivars to cope with projected climate scenarios.

## 1. Introduction

Wheat is one of the most important cereal crops in terms of production and area harvested at a global level, with over 734 million tonnes of annual production in 2018 [1], and this is expected to rise as the world’s demand expands into the future [2]. Alongside this challenge is agriculture’s need to cope with the effects of climate change (CC) to ensure the food security of a growing population. Atmospheric concentrations of CO_2_ ([CO_2_]) have increased since the pre-industrial era from ≈300 ppm to 410 ppm and are predicted to continue rising to between 800 and 1060 ppm by the end of this century [3]. The responses of C3 plants, such as wheat, to *e*[CO_2_] are well documented. These plants have shown enhanced *An* and reduced stomatal conductance (*gs*) [4,5,6,7], which is often associated with increases in biomass [5,8]. However, the majority of experiments have been carried out in greenhouses or “chambers” to study the effects of *e*[CO_2_], and they are known to have several limitations. In contrast, large-scale free-air CO_2_ enrichment (FACE) technology has advantages, i.e., it does not use confinement structures and the plants undergo development in natural and open-air conditions [9].

A meta-analysis using FACE technology showed that stimulation of the photosynthetic rate is about 30%; whereas, the stomatal conductance is reduced on average by 20%. Additionally, the *V_c_*_,*max*_ and the *J_max_* were reduced in C3 plants at *e*[CO_2_], although the reduction in *V_c_*_,*max*_ was more pronounced than in *J_max_* [4]. In fact, these modifications in gas exchange affect both C and N dynamics in plants. The current gains in C acquisition often result in lower N uptake and decreased N concentration, which are possibly due to dilution of N by excess carbohydrate accumulation or increased growth at high [CO_2_] [10]. Although an increase in the nitrogen use efficiency (NUE) in C3 plants has been reported [6], N assimilation into amino acids and proteins can be inhibited under rising [CO_2_] [11,12,13]. Nevertheless, these changes depend on the environmental conditions, plant groups, and developmental stage at which the measurements are performed [14].

In this sense, a previous study conducted under greenhouse conditions at two developmental stages (vegetative and post-anthesis) in wheat plants exposed to *e*[CO_2_] [14], reported that a higher *An* was higher under *e*[CO_2_], particularly at the vegetative stage, and a progressive decline in N content in leaves from the vegetative to post-anthesis stages. In FACE conditions, a decrease in the N concentration in leaves under *e*[CO_2_] before anthesis was observed in wheat, which was due to increased growth of plants under such conditions [8]. In addition, a number of studies have compared the responses of organs of wheat plants at *e*[CO_2_], such as leaves and spikes, but these have been conducted under greenhouses conditions. Li et al. [7] noted that both the leaf and spike C concentrations were enhanced by high CO_2_ while the N concentration was lower in leaves but was unaffected by *e*[CO_2_] in the spike. Similar results were found by Aranjuelo et al. [12] where *e*[CO_2_] had no effect on N content in the ear. 

Taking this into account, few studies have simultaneously studied the responses of diverse organs at different crop developmental stages using FACE environments. Consequently, the main objective of this study was to evaluate the effect of *e*[CO_2_] in both agronomic and physiological responses, as well as C and N metabolism at three developmental stages (vegetative, anthesis, and maturity) in two cultivars of wheat (Norin and Triumph) using the FACE technology.

## 2. Results

### 2.1. Agronomic and Physiological Parameters

The *e*[CO_2_] tended to increase the grain dry matter (DM), straw DM, thousand-kernel weight (TKW), and harvest index (HI) in both of the cultivars of wheat evaluated, although no significant differences were found in *e*[CO_2_] with respect to *a*[CO_2_] (Figure 1). In particular, the highest increase under *e*[CO_2_] was observed in the grain DM, with 11% for Triumph and 8% for Norin. Furthermore, significant differences were observed between cultivars in the agronomic parameters, except for grain DM, where Triumph showed higher values in straw DM and TKW than Norin, but lower in HI.

Regarding the physiological parameters, gas exchange parameters such as *An*, *V_c_*_,*max*_, and *J_max_* were significantly affected by *e*[CO_2_] (Figure 2). *An* was increased by 36% and 24% in Triumph and Norin, respectively. In contrast, the *V_c_*_,*max*_ and *J_max_* decreased under *e*[CO_2_] conditions, but while the percentage of decrease in *V_c_*_,*max*_ was similar in both Triumph (40%) and Norin (48%), the reduction in *J_max_* was more pronounced in Triumph than Norin, at 24% and 4%, respectively. Also, the results of the two-way ANOVA indicated that both [CO_2_] and cultivar factors had a significant effect on gas exchange parameters, except for stomatal conductance, whereas the interaction had no effect.

### 2.2. C and N Compounds at Different Stages of Development 

Parameters such as starch content, N, and free total amino acids were analyzed at three stages of development (elongation, anthesis, and maturation) and in different organs of plants grown under *a*[CO_2_] and *e*[CO^2^].

#### 2.2.1. Elongation Stage

At the elongation stage, leaves and stems were evaluated in Triumph and Norin (Figure 3), with differential behavior observed when they were grown under *e*[CO_2_] conditions. In Triumph (Figure 3A) the parameters tended to decrease in both leaves and stems under *e*[CO_2_] compared to *a*[CO_2_]. Nonetheless, significant differences between organs were only observed in the free amino acid content, for which the stems showed a pronounced content reduction (≈60%) relative to leaves (10%). On other hand, Norin showed a reduction in starch content (Figure 3B) in both leaves and stems at *e*[CO_2_], but the N and free amino acids had higher values under such conditions, with 5% and 8% average increases, respectively. However, no significant differences were observed between the organs evaluated.

#### 2.2.2. Anthesis Stage

The leaf, stem, and ear organs were analyzed at the anthesis stage (Figure 4). In Triumph, the starch content varied among organs, with the leaves showing a reduction of 52% relative to *a*[CO_2_], but stems and ears increasing by 7% and 163%, respectively. Also, no significant differences were observed for N or the free amino acid content, although greater increases in amino acids were observed in *e*[CO_2_] for the three organs evaluated. In Norin (Figure 4B), similar results were obtained at the elongation and anthesis stages, with reductions in the average of starch content (49%) in the three organs and at both stages (see above), but the increases in N (14%) and amino acid (82%) contents were more pronounced at the anthesis stage than the elongation stage. However, significant differences among organs were detected in the amino acid content, for which the leaf and stem values increased by around 100% while the increase in ears was only 28% under *e*[CO_2_]. 

#### 2.2.3. Maturation Stage

At the maturation stage, the leaf, stem, peduncle, shell, and grain organs were evaluated (Figure 5). In Triumph (Figure 5A) grown under *e*[CO_2_], there were significant differences in starch among the organs observed, with reductions in the leaf content, as well as reductions observed in the elongation and anthesis stages, but increases in stems (64%) and peduncles (139%). In contrast, the starch content in the shells (close to the grain) and the grain decreased by 13% and 2%, respectively. Moreover, a general increase was observed in N and amino acid content under *e*[CO_2_], except in shells and grain, which showed decreases in amino acid content of 5% and 30%, respectively, although no significant differences were observed. In Norin, a significant decline was detected in starch content in the majority of the organs evaluated under *e*[CO_2_] (Figure 5B). In terms of starch content, shells in particular presented the greatest reductions (75%). In addition, there was an increase in N content in the leaves (25%), stems (25%), and peduncles (5%) under *e*[CO_2_], but N decreased in the shells (33%) and grain (4%). Similarly, the amino acid content increased in leaves (27%) and peduncles (24%), but it decreased significantly in shells (61%) and grain (7%). 

## 3. Discussion


**Agronomic and physiologic characterization.**


Wheat performance has been reviewed and evaluated under *e*[CO_2_] conditions in a number of FACE experiments [6,8,9,15,16,17,18]. A meta-analysis [15] has also evaluated the yield response in 18 crops under FACE conditions, including wheat, where exposure of wheat plants to *e*[CO_2_] increased the yield by almost 14% with respect to *a*[CO_2_]. In general terms, our study did not show a clear *e*[CO_2_] effect on wheat yield. Similar results were found by Broberg et al. [19] and Tcherkez et al. [18], who indicated that the absence of significant differences in wheat yield parameters is determined by each cultivar, the crop management protocols used, and environmental factors such as water availability. However, genetics could have a strong effect in the absence of differences in agronomic parameters, especially in this study, in which the difference between *a*[CO_2_] and *e*[CO_2_] was low, which could have led to a minimal effect on the yield parameters [18]. Additionally, these results could be related to the lack of significant effects of *e*[CO_2_] on *gs* in this study, considering that the yield and plant growth is driven by the CO_2_ uptake [13]. The initial stimulation of *An* under *e*[CO_2_] could have been caused by an increase in CO_2_ around the site of CO_2_ fixation [5,6]. The significant increase in *An* and decrease in *V_c,max_* and *J_max_*, contrasting with the slight change in plant, DM could be associated with a phenomenon known as photosynthetic acclimation, which is characterized by the suppression of different routes mainly associated with N assimilation [5] and further explained below. Similar to our findings, other studies have observed increases in *An* of close to 20% in wheat plants evaluated with the FACE methodology [18,20].

**Changes in the C and N dynamics under *e*[CO_2_] according to the cultivar, organ, and phenological stage evaluated**. 

In our study, the effect of *e*[CO_2_] resulted in a differential responses between cultivars, with changes in C and N dynamics in different organs and at three phenological stages (elongation, anthesis, and maturation) in wheat plants. 

A previous study [18] has shown that photosynthetic down-regulation under *e*[CO_2_] is conditioned by the plant’s ability to develop new sinks (e.g., new vegetative or reproductive structures, such as ears), or to expand the storage capacity or growth rate of existing sinks. Developing strong C sinks would contribute to overcoming potential increases in the carbohydrate balance of the leaves that could induce the acclimation of photosynthetic machinery. Within this context, it should be also noted that the high [CO_2_] has been described to alter the source–sink ratio because an excess or deprivation in the C and N supply could restrict the development of new sink organs, which exacerbates the source–sink imbalance in plants grown at *e*[CO_2_] [9,21]. In relation to the identification of potential C sinks, the C components can be transferred to green tissue, redistributed from reserves, or stored in vegetative organs such as stems [12,21]. Indeed, C compounds, such as starch content, have been observed to undergo degradation for the purpose of synthetizing sucrose and exporting it directly to sink organs [22]. In our study, this effect has been observed mainly at the elongation stage in both cultivars and then a decrease in the last stages (anthesis and maturation) was observed in Norin. By contrast, Triumph showed a significant increase in starch content during anthesis and maturation in stems, ears, peduncles, and shells, possibly as a strategy to supply sink organs [5].

In this context, the behavior of the Triumph cultivar could be associated with acclimation of the photosynthetic capacity of the plants to high [CO_2_], as reported by Ainsworth and Long [9]. The acclimation process could cause the accumulation of sugars that exceed the capacity of the plant to produce new sink organs, and thus the plants decrease their photosynthetic rate and tend to form storage compounds [21]. Similar results in wheat plants exposed to *e*[CO_2_] were found by Aranjuelo et al. [12], who indicated that increases in starch are related to alterations in C utilization that favor the storage of C. Moreover, during the grain-filling stage, the C demand by sink organs (grain) is usually supplied by ear photosynthesis [23] and also by translocation from stems [12]. Taking this into account, our results showed that Triumph had greater accumulation of starch in stems and peduncles, and this suggests that these organs might translocate C compounds to sink organs such as the grain and shells, in which the starch content tended to decrease under *e*[CO_2_]. 

Furthermore, C is tightly related to the N cycle, and frequently, N also limits plant growth [24]. Further, it should be also observed that *e*[CO_2_] might modify the leaf biochemistry and grain quality throughout the phenological stages. During anthesis there is a detrimental effect on the photosynthetic machinery of flag leaves undergoing a more rapid decline in N due to premature senescence, which can then impact grain filling during maturation [18]. Within this context, the current study revealed that, in general, Triumph showed a N decrease during the elongation and anthesis stages in almost all the organs evaluated, which can be explained by the CO_2_ enrichment causing a decrease in N content due to a dilution effect following the increase in biomass [17,24] or due to a decrease in the allocation of N to all the leaves at the whole-plant level [5]. However, in our study, the increase in biomass was not significant, which could indicate a minimal N dilution effect. This suggests that the reduction in N content, especially in Triumph, may be due to a reduction in N demand [25], repression of photorespiration [11,13], or the allocation of N to sink organs such as roots or other organs closer to the grain [5]. Moreover, the decline in N in Triumph, in particular, may be associated with photosynthesis acclimation, as reported in previous studies with wheat plants at high CO_2_ [10,12,17], and also supported by the accumulation of starch detected in some organs. Additionally, the exposure of plants to *e*[CO_2_] could promote the N assimilation, possibly in the form of amino acids, as mentioned by Adavi and Sathee [11]; for this, the amino acids may be increased, as observed in this study, in both cultivars. Shell and grain organs showed a decrease in amino acid content, which indicates that there is a minimal translocation from leaves, stems and peduncles. These differential responses between the cultivars evaluated in this study are possibly due to genetics, the ability to respond to new conditions, the availability of nutrients in the soil, and environmental conditions [21]. In this sense, the Triumph cultivar showed a higher photosynthetic acclimation to high [CO_2_] than Norin, which was confirmed by the accumulation of starch and a decrease in N content in the majority of organs examined.

## 4. Materials and Methods

### 4.1. Plant Growth and Experimental Conditions

The experiment was conducted in the mini free-air carbon dioxide enrichment system of the Chinese Academy of Agricultural Sciences (CAAS-FACE system) in Changping (40°10′ N, 116°14′ E), Beijing, China, from 2016 to 2017. The mean rainfall and temperature during the wheat growth period were 203 mm and 8 °C, respectively. The Mini-FACE system consisted of 12 experimental plots, including 6 elevated *e*[CO_2_] rings (550 ± 17 μmol mol^−1^) and 6 *a*[CO_2_] rings (415 ± 16 μmol mol^−1^), each with a diameter of 4 m. The experimental plots were at least 14 m apart to minimize cross-contamination of CO_2_ between the experimental treatments [26] (Appendix A). 

Two winter wheat (*Triticum aestivum* L.) cultivars, Triumph and Norin, were selected for this study. Both wheat varieties were sown in each of the CO_2_ treatment plots at the same time, with a plot area of 3.75 m^2^. The planting density was 333 plants per square meter and the row interval was 20 cm under both the *e*[CO_2_] rings (∼550 μmol mol^−1^) and *a*[CO_2_] rings (∼415 μmol mol^−1^), with 3 replicates per treatment. The varieties were planted randomly in each plot to minimize the effects of soil variation. 

In the experiment, the soil was classified as clay loam with a pH of 8.4. Fertilizer application comprised granular urea (N, 46%), diammonium phosphate (N:P_2_O_5_ 13:44%), and potassium chloride (K_2_O, 60%), which were applied as basal fertilizers at the rates of 100 kg ha^−2^, 165 kg ha^−2^ and 90 kg ha^−2^, respectively. At the jointing stage, granular urea was applied as a side dressing at a rate of 100 kg ha^−2^ on 28 April 2017. Irrigation was applied twice during the entire winter wheat growing season: the wintering irrigation was applied at a rate of 750 m^3^ ha^−2^ on 23 November 2016, and the spring irrigation was applied at a rate of 750 m^3^ ha^−2^ at the jointing stage after side-dressing fertilization.

### 4.2. Measurements of Agronomic and Gas Exchange Parameters

The agronomic parameters were determined from a 1 m^2^ patch inside each ring. The crop areas chosen were destructively sampled at the ripening stage of the plants. The samples were dried at 60 °C for 48 h and then weighed for the calculation of grain and straw DM. Furthermore, TKW and HI (HI = ratio between seed weight and total DM) were determined. 

Regarding gas exchange parameters, analyses were carried out in healthy expanded flag leaves grown under *a*[CO_2_] and *e*[CO_2_] conditions, selected randomly in the middle of each ring. The *An* and *gs* were estimated using a portable gas exchange photosynthesis system (Li-Cor 6400, Lincoln, NE, USA) with a photosynthetic photon flux density (PPFD) of 1500–1600 µmol m^−2^ s^−1^. An external light source composed of an LED lamp was supplied to achieve the PPFD inside the chamber. The vapor pressure deficit was 1.5 kPa. CO_2_ response curves were constructed from a series of measurements whereby photosynthesis was determined first under standard conditions (400 µmol mol^−1^), then at low CO_2_, then back to standard conditions, and then at high overall CO_2_, with the sequence of CO_2_ concentrations in the reference channel being 400, 200, 100, 50, 200, 400, 600, 750, and 950 µmol mol^−1^. The values of *An* and sub-stomatal CO_2_ mole fraction (*Ci*) were used to estimate the *V_c_*_,*max*_ and *J_max_* contribution to ribulose 1,5-bisphosphate regeneration (measurements made in Germany) according to the method of Harley et al. [27].

### 4.3. C and N Content

Leaf, stem, ear, peduncle, shell, and grain samples were collected at different developmental stages, dried at 60 °C for 48 h, and then ground. Next, 1.5 mg of the sample were weighed to determine the C and N content using an isotope ratio mass spectrometer (Delta C; Finnigan, Mat., Bremen, Germany) coupled to an elemental analyzer (EA1108; Carlo Erba Instrumentazione, Milan, Italy) operating in continuous flow mode.

### 4.4. Starch Content

Dried plant material from the different organs mentioned above was used to quantify the starch. The samples were ground to a fine powder, weighed close to 25 mg and suspended in ethanol (80%) in Eppendorf tubes, shaken in a thermomixer (90 min, 70 °C, 1100 rpm) and centrifuged (10 min, 22 °C, 20,800× *g*). For starch solubilization, the pellet was resuspended in 0.2N KOH and the starch was extracted using an amyloglucosidase test kit (R-Biopharm AG, Darmstadt, Germany). Finally, starch quantification was performed through absorbance measurements at 340 nm. 

### 4.5. Free Total Amino Acid Content

The total amino acid content was determined on dried material of different organs. The samples were ground and weighed close to 20 mg. For the extraction of free amino acids, the plant tissues were homogenized in 80% ethanol, and then they were incubated in a thermomixer (60 min, 80 °C, 600 rpm) and centrifuged (10 min, 4 °C, 14,000× *g*). The supernatant was completely dehydrated on speed vacuum for 3 h and the pellet was resuspended in 100 µL of milli-Q water. The amino acid content was determined by high- performance liquid chromatography (Water Corporation) after derivatization with an ACCQ-Fluor™ Reagent kit (Waters, Milford, MA, USA) based on borate buffer, acetonitrile, and AQC derivatizing reagent (6-aminoquinolyl-N-hydroxysuccinimidyl carbamate). 

### 4.6. Statistical Analysis

Data analyses were conducted using a factorial design with two-way ANOVA (RStudio^®^ v.3.4.2, 2017; Boston, MA, USA). The factors used were the cultivars (Triumph and Norin) and [CO_2_] (ambient and elevated). When the main effect of each factor and/or the interaction between these was significant, the Tukey HSD test was applied to compare means. Significant differences were considered when *p* < 0.05. In the case of the parameters C, N, starch, and free total amino acids, the percentage of change under *e*[CO_2_] with respect to *a*[CO_2_] was calculated.

## 5. Conclusions

The effect of *e*[CO_2_] on biomass and agronomic parameters was minimal in both cultivars evaluated, which could be associated with the small difference between the two [CO_2_] conditions. However, in terms of physiology and C and N dynamics, the two cultivars had differential responses to high [CO_2_] under FACE conditions. Despite the initial increase in the photosynthetic rate and the decrease in *V_c_*_,*max*_ and *J_max_* in both cultivars, an accumulation of starch and a decline in N content was observed in Triumph in the majority of organs, and these factors were associated with photosynthetic acclimation. Conversely, Norin showed a decrease in starch and an increase in N content. At the organ level, the shells and grain had decreased N and amino acid content, contrary to the leaves, stems, and peduncles, and this indicated a minimal translocation from source to sink organs and demonstrated that *e*[CO_2_] might modify the source–sink relationship between organs. Finally, other FACE experiments will be required to test genetic variation and its particular adaptation strategies in response to *e*[CO_2_].

## Figures and Tables

**Figure 1 plants-12-00686-f001:**
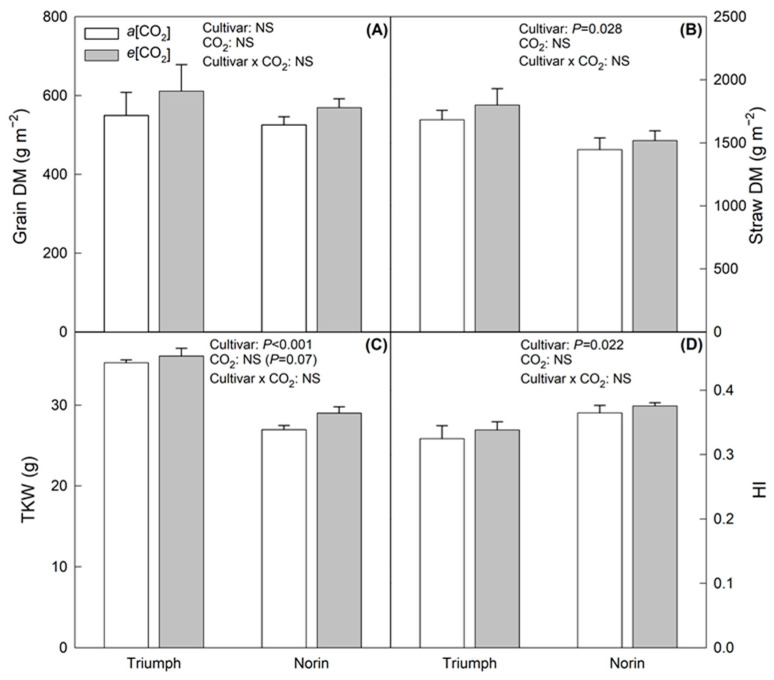
Agronomic responses to *e*[CO_2_] in both cultivars evaluated (Triumph and Norin). (**A**) Grain DM; (**B**) Straw DM; (**C**) TKW; and (**D**) HI. Each bar indicates standard error of means (*n* = 3) and cultivar, CO_2_ and cultivar x CO_2_ represent the ANOVA results of cultivar, [CO_2_] and the interaction between them.

**Figure 2 plants-12-00686-f002:**
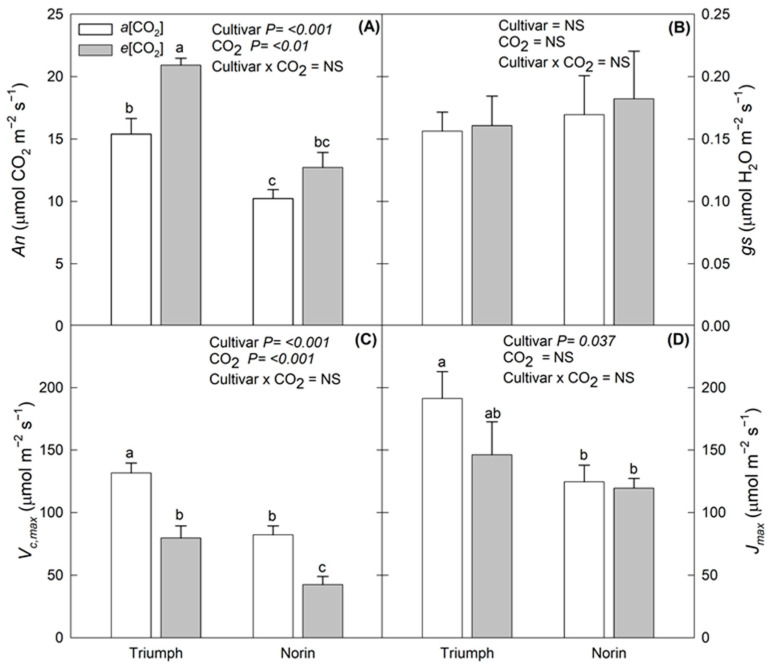
Physiological responses to *e*[CO_2_] in both cultivars evaluated (Triumph and Norin). (**A**) *An*; (**B**) *gs*; (**C**) *V_c_*_,*max*_; and (**D**) *J_max_*. Each bar indicates standard error of means (*n* = 3) and cultivar, CO_2_ and cultivar x CO_2_ represents the ANOVA results of cultivar, [CO_2_] and the interaction between them. Different letters indicate significant differences among treatments and cultivar.

**Figure 3 plants-12-00686-f003:**
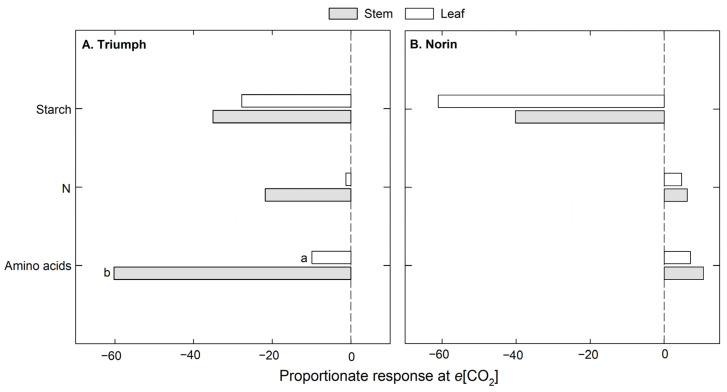
Percentage (%) of change under *e*[CO_2_] with respect to *a*[CO_2_] in two organs evaluated (stems and leaves) at the elongation stage in both wheat cultivars: (**A**) Triumph and (**B**) Norin. The letters indicate significant differences between organs (*p* < 0.05).

**Figure 4 plants-12-00686-f004:**
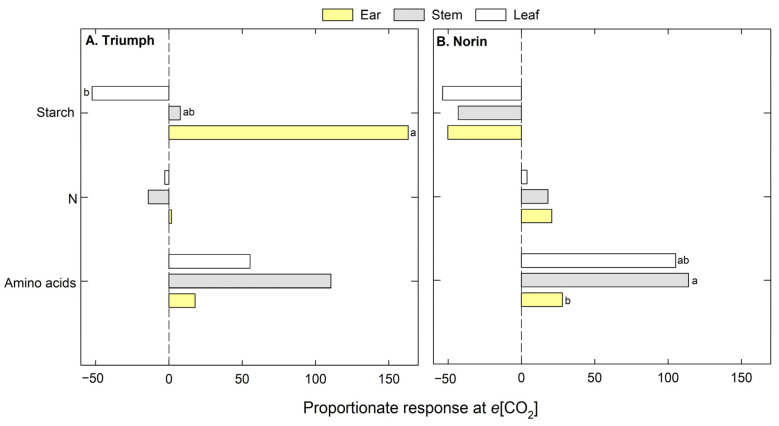
Percentage (%) of change under *e*[CO_2_] relative to *a*[CO_2_] in three organs evaluated (stems, leaves, and ears) at anthesis in both wheat cultivars: (**A**) Triumph and (**B**) Norin. The letters indicate significant differences between organs (*p* < 0.05).

**Figure 5 plants-12-00686-f005:**
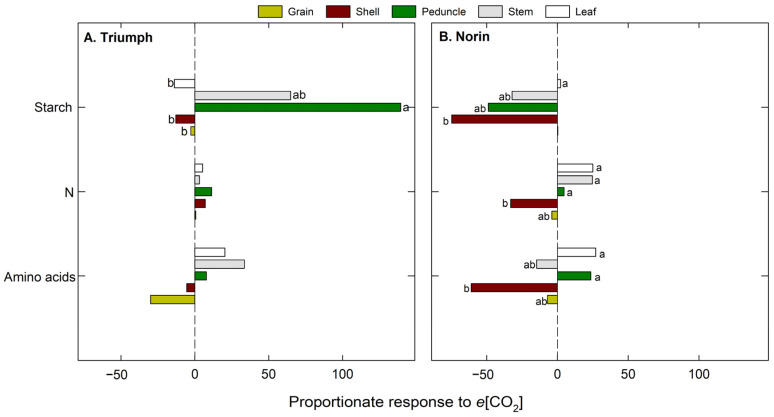
Percentage (%) of change under *e*[CO_2_] relative to *a*[CO_2_] in five organs evaluated (stems, leaves, peduncles, shells, and grain) at maturation in both wheat cultivars: (**A**) Triumph and (**B**) Norin. The letters indicate significant differences between organs (*p* < 0.05).

## Data Availability

The raw data supporting the conclusions of this article will be made available by the authors without undue reservation.

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
