# Peer review of "Differential Effect of Free-Air CO2 Enrichment (FACE) in Different Organs and Growth Stages of Two Cultivars of Durum Wheat"

_plants, 2023, doi:10.3390/plants12030686_

Round 1

Reviewer 1 Report

Differential responses between the cultivars evaluated in your study are due possibly to genetics, the ability to respond to new conditions availability of nutrients in soil environmental conditions. 

Author Response

Differential responses between the cultivars evaluated in your study are due possibly to genetics, the ability to respond to new conditions availability of nutrients in soil environmental conditions. In agreement with the comment made by the reviewer, in the new version of the manuscript (Lines 219-220), we have added a sentence remarking that the differences can be associated to the genetic variability of analyzed cultivars.

Reviewer 2 Report

There are only results in abstract (no background, no methods, no summarising sentence about future research, what the mean in numbers of elevated CO2, minimal effect of e[CO2] compared with what?).

Line 31: "of CO2] has increased 31"

Line 41: "photosynthetic 41 rate is ca. 30%" - what is ca.?

In instructions for Authors: "should be defined the first time they appear in each of three sections: the abstract; the main text; the first figure or table." But there are no explanations in introduction: An (line 34), Vc,max and the Jmax (line 43)

[CO2]- why are used [ ]?

What was the soil systematic unit / soil type and main agrochemical properties of the soil?

The scheme of experiment / layout of plots would help to better understanding of field experiment.

How was counted HI? What sample size was used?

No information about sampling strategy, please supplement.

What mean the bars in Figure 1 and 2?

In Conclusions a sentence about future research is needed.

Author Response

  1. There are only results in abstract (no background, no methods, no summarising sentence about future research, what the mean in numbers of elevated CO2, minimal effect of e[CO2] compared with what?). The abstract has been improved accordingly (Page 1).
  2. Line 31: "of CO2] has increased 31". The word was corrected according to the comment made by the referee (Line 39).
  3. Line 41: "photosynthetic 41 rate is ca. 30%" - what is ca.? Ca would be the abbreviation of “close at”. In order to make the text more understandable, Ca was replaced in throughout the whole text.
  4. In instructions for Authors: "should be defined the first time they appear in each of three sections: the abstract; the main text; the first figure or table." But there are no explanations in introduction: An (line 34), Vc,max and the Jmax (line 43). In agreement with the comment made, An, Vcmax and Jmax have been defined in the new version of the abstract.
  5. [CO2]- why are used [ ]? It is used to indicate the concentration of CO2. This aspect has been defined in the introduction section (Line 39) of the manuscript.
  6. What was the soil systematic unit / soil type and main agrochemical properties of the soil? As requested by referee, this information was included in Material and Methods section with the available information (Lines 410-411).
  7. The scheme of experiment / layout of plots would help to better understanding of field experiment. In order to more clearly explain the experimental design of the study, a supplementary figure (Fig. S1) has been added to the manuscript,
  8. How was counted HI? What sample size was used? Following the recommendation made by the referee, this information was explained in Materials and methods section (Lines 422-423).
  9. No information about sampling strategy, please supplement. The information was supplemented in Materials and methods section (Lines 425-426) of the revised version.
  10. What mean the bars in Figure 1 and 2? It has been clarified below each figure.
  11. In Conclusions a sentence about future research is needed. In agreement with the comment made by the referee, a reference to future perspectives was included in conclusion section (Lines 489:491).

Reviewer 3 Report

The paper deals with agronomic and physiological wheat responses in plots arranged within a mini-FACE  experiment, in which two durum wheat cultivars were grown at ambient and elevated [CO2] in Changping, Beijing, China.

As for classical agronomic traits (grain yield, straw, TKW and HI) Authors' statistical analyses do not report significant effects of the contrasting CO2 regimes. Conversely, the photosinthetic traits An, Vcmax, Jmax - but not gs - show significant differences between the two [CO2] environments, with no interaction with wheat varieties.

Several results reported by Authors' appear not so solid, at least in statistical terms. In my humble opinion, this is intrinsically linked to the limited range of the two [CO2] environments tested in the mini-FACE apparatus. Of course, the experiment should have taken into account that in advance, possibly arranging a larger number of replications, but it is understandable that those experimental conditions have been fixed and/or not easy to customize.

As for the layout of the paper, Fig. 1 must be enlarged, tentatively to the size of Fig. 2, as in the present form details are unreadable. Also Fig. 3 and 4 have different sizes. I am wondering whether it would be useful to present results about delta 13C separately in a new table, so that the remaining traits in Fig. 3, 4 and 5  could increase size and gain more visibility.

Mostly in Discussion, I found a few problems and/or typos in the English text. See for instance lines 152, 178, 200. A typo is found also in the Fig. 3 description (steam for stem). I found also a couple of English problems in lines 55 and 61 of the Introduction.

In Mat. & Meth., I do not understand the meaning of the sentence in line 240 and 241. I am also not familiar with the units of fertilizers supply (kg hm-2). Do you mean kg ha-1? More importantly, I would suggest to expand the information about gas exchange parameters, as in the present form they could not be sufficient to completely satisfy readers' without consulting the literature. How many leaves were measured per each experimental unit?

Finally, in the Discussion (as well in the Conclusions), Authors could have spend more words to explain the observed differences in the behavior (and consequently in acclimation) of the two durum varieties. Are there any differences in plant structure, HI and/or year of release of those genotypes?

Author Response

  1. As for classical agronomic traits (grain yield, straw, TKW and HI) Authors' statistical analyses do not report significant effects of the contrasting CO2 Conversely, the photosinthetic traits An, Vcmax, Jmax - but not gs - show significant differences between the two [CO2] environments, with no interaction with wheat varieties. Several results reported by Authors' appear not so solid, at least in statistical terms. In my humble opinion, this is intrinsically linked to the limited range of the two [CO2] environments tested in the mini-FACE apparatus. Of course, the experiment should have taken into account that in advance, possibly arranging a larger number of replications, but it is understandable that those experimental conditions have been fixed and/or not easy to customize. In agreement with the comment made by the referee, we would also like to remark that it is difficult to modify the experimental conditions in the FACE facilities. Within this context, in the new version of the paper include the possibility that the low difference between two [CO2] environments (415 versus 550 µmol mol-1 CO2) in the limited differences in agronomical parameters of both cultivars (Lines 220-221).
  2. As for the layout of the paper, Fig. 1 must be enlarged, tentatively to the size of Fig. 2, as in the present form details are unreadable. Also Fig. 3 and 4 have different sizes. I am wondering whether it would be useful to present results about delta 13C separately in a new table, so that the remaining traits in Fig. 3, 4 and 5 could increase size and gain more visibility. Following the recommendation made by the referee, the figure 1 was enlarged to size of figure 2 and also. Regarding, the 13C isotopic composition data, after a critical discussion between the all the authors, we consider that, the information given by this parameter might be redundant and does not provide novel information to the manuscript. For this reason, those data have been removed from the text.
  3. Mostly in Discussion, I found a few problems and/or typos in the English text. See for instance lines 152, 178, 200. A typo is found also in the Fig. 3 description (steam for stem). I found also a couple of English problems in lines 55 and 61 of the Introduction. The text has been revised and corrected.
  4. In Mat. & Meth., I do not understand the meaning of the sentence in line 240 and 241. I am also not familiar with the units of fertilizers supply (kg hm-2). Do you mean kg ha-1? More importantly, I would suggest to expand the information about gas exchange parameters, as in the present form they could not be sufficient to completely satisfy readers' without consulting the literature. How many leaves were measured per each experimental unit? In relation to the comment made by the reviewer, in the revised version of the manuscript we have revised fertilization units, that, as observed, refer to hectares (ha). Regarding gas exchange analyses, the procedure followed to conduct such determinations have been explained in more detail.
  5. Finally, in the Discussion (as well in the Conclusions), Authors could have spend more words to explain the observed differences in the behavior (and consequently in acclimation) of the two durum varieties. Are there any differences in plant structure, HI and/or year of release of those genotypes? In relation to the comment made, we would like to observe that, both cultivars showed a similar phenotype. Although in case of Norin, HI was higher than that in Triumph, plant structure was similar in both cultivars. Regarding the differential response of Norin and Triumph, the Discussion and Conclusion sections have been revised so to remark this point (Lines 223:228, 237:325 and 482:485).
